# On the Stability of Fine-tuning BERT: Misconceptions, Explanations, and Strong Baselines

**Marius Mosbach**
Spoken Language Systems (LSV)
Saarland Informatics Campus, Saarland University
mmosbach@lsv.uni-saarland.de

**Maksym Andriushchenko**
Theory of Machine Learning Lab
École polytechnique fédérale de Lausanne
maksym.andriushchenko@epfl.ch

**Dietrich Klakow**
Spoken Language Systems (LSV)
Saarland Informatics Campus, Saarland University
dietrich.klakow@lsv.uni-saarland.de

## ABSTRACT

Fine-tuning pre-trained transformer-based language models such as BERT has become a common practice dominating leaderboards across various NLP benchmarks. Despite the strong empirical performance of fine-tuned models, fine-tuning is an unstable process: training the same model with multiple random seeds can result in a large variance of the task performance. Previous literature (Devlin et al., 2019; Lee et al., 2020; Dodge et al., 2020) identified two potential reasons for the observed instability: catastrophic forgetting and small size of the fine-tuning datasets. In this paper, we show that both hypotheses fail to explain the fine-tuning instability. We analyze BERT, RoBERTa, and ALBERT, fine-tuned on commonly used datasets from the GLUE benchmark, and show that the observed instability is caused by optimization difficulties that lead to vanishing gradients. Additionally, we show that the remaining variance of the downstream task performance can be attributed to differences in generalization where fine-tuned models with the same training loss exhibit noticeably different test performance. Based on our analysis, we present a simple but strong baseline that makes fine-tuning BERT-based models significantly more stable than the previously proposed approaches. Code to reproduce our results is available online: https://github.com/uds-lsv/bert-stable-fine-tuning.

## 1 INTRODUCTION

Pre-trained transformer-based masked language models such as BERT (Devlin et al., 2019), RoBERTa (Liu et al., 2019), and ALBERT (Lan et al., 2020) have had a dramatic impact on the NLP landscape in the recent year. The standard recipe for using such models typically involves training a pre-trained model for a few epochs on a supervised downstream dataset, which is known as *fine-tuning*. While fine-tuning has led to impressive empirical results, dominating a large variety of English NLP benchmarks such as GLUE (Wang et al., 2019b) and SuperGLUE (Wang et al., 2019a), it is still poorly understood. Not only have fine-tuned models been shown to pick up spurious patterns and biases present in the training data (Niven and Kao, 2019; McCoy et al., 2019), but also to exhibit a large training *instability*: fine-tuning a model multiple times on the same dataset, varying only the random seed, leads to a large standard deviation of the fine-tuning accuracy (Devlin et al., 2019; Dodge et al., 2020).

Few methods have been proposed to solve the observed instability (Phang et al., 2018; Lee et al., 2020), however without providing a sufficient understanding of why fine-tuning is prone to such failure. The goal of this work is to address this shortcoming. More specifically, we investigate the following question:

*Why is fine-tuning prone to failures and how can we improve its stability?*

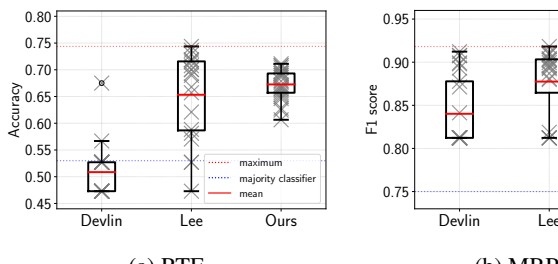 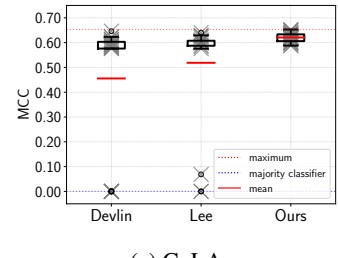

(a) RTE           (b) MRPC           (c) CoLA

Figure 1: Our proposed fine-tuning strategy leads to very stable results with very concentrated development set performance over 25 different random seeds across all three datasets on BERT. In particular, we significantly outperform the recently proposed approach of Lee et al. (2020) in terms of fine-tuning stability.

We start by investigating two common hypotheses for fine-tuning instability: catastrophic forgetting and small size of the fine-tuning datasets and demonstrate that both hypotheses fail to explain fine-tuning instability. We then investigate fine-tuning failures on datasets from the popular GLUE benchmark and show that the observed fine-tuning instability can be decomposed into two separate aspects: (1) optimization difficulties early in training, characterized by vanishing gradients, and (2) differences in generalization late in training, characterized by a large variance of development set accuracy for runs with almost equivalent training loss.

Based on our analysis, we present a simple but strong baseline for fine-tuning pre-trained language models that significantly improves the fine-tuning stability compared to previous works (Fig. 1). Moreover, we show that our findings apply not only to the widely used BERT model but also to more recent pre-trained models such as RoBERTa and ALBERT.

## 2   RELATED WORK

The fine-tuning instability of BERT has been pointed out in various studies. Devlin et al. (2019) report instabilities when fine-tuning BERT$_{\text{LARGE}}$ on small datasets and resort to performing multiple restarts of fine-tuning and selecting the model that performs best on the development set. Recently, Dodge et al. (2020) performed a large-scale empirical investigation of the fine-tuning instability of BERT. They found dramatic variations in fine-tuning accuracy across multiple restarts and argue how it might be related to the choice of random seed and the dataset size.

Few approaches have been proposed to directly address the observed fine-tuning instability. Phang et al. (2018) study intermediate task training (STILTS) before fine-tuning with the goal of improving performance on the GLUE benchmark. They also find that their proposed method leads to improved fine-tuning stability. However, due to the intermediate task training, their work is not directly comparable to ours. Lee et al. (2020) propose a new regularization technique termed Mixout. The authors show that Mixout improves stability during fine-tuning which they attribute to the prevention of catastrophic forgetting.

Another line of work investigates optimization difficulties of *pre-training* transformer-based language models (Xiong et al., 2020; Liu et al., 2020). Similar to our work, they highlight the importance of the learning rate warmup for optimization. Both works focus on pre-training and we hence view them as orthogonal to our work.

## 3   BACKGROUND

### 3.1   DATASETS

We study four datasets from the GLUE benchmark (Wang et al., 2019b) following previous work studying instability during fine-tuning: CoLA, MRPC, RTE, and QNLI. Detailed statistics for each of the datasets can be found in Section 7.2 in the Appendix.

**CoLA.** The Corpus of Linguistic Acceptability (Warstadt et al., 2018) is a sentence-level classification task containing sentences labeled as either grammatical or ungrammatical. Fine-tuning on CoLA was observed to be particularly stable in previous work (Phang et al., 2018; Dodge et al., 2020; Lee et al., 2020). Performance on CoLA is reported in Matthew's correlation coefficient (MCC).

**MRPC.** The Microsoft Research Paraphrase Corpus (Dolan and Brockett, 2005) is a sentence-pair classification task. Given two sentences, a model has to judge whether the sentences paraphrases of each other. Performance on MRPC is measured using the $F_1$ score.

**RTE.** The Recognizing Textual Entailment dataset is a collection of sentence-pairs collected from a series of textual entailment challenges (Dagan et al., 2005; Bar-Haim et al., 2006; Giampiccolo et al., 2007; Bentivogli et al., 2009). RTE is the second smallest dataset in the GLUE benchmark and fine-tuning on RTE was observed to be particularly unstable (Phang et al., 2018; Dodge et al., 2020; Lee et al., 2020). Accuracy is used to measure performance on RTE.

**QNLI.** The Question-answering Natural Language Inference dataset contains sentence pairs obtained from SQuAD (Rajpurkar et al., 2016). Wang et al. (2019b) converted SQuAD into a sentence pair classification task by forming a pair between each question and each sentence in the corresponding paragraph. The task is to determine whether the context sentence contains the answer to the question, i.e. entails the answer. Accuracy is used to measure performance on QNLI.

## 3.2 FINE-TUNING

Unless mentioned otherwise, we follow the default fine-tuning strategy recommended by Devlin et al. (2019): we fine-tune uncased BERT$_{\text{LARGE}}$ (henceforth BERT) using a batch size of 16 and a learning rate of $2e-5$. The learning rate is linearly increased from $0$ to $2e-5$ for the first $10\%$ of iterations—which is known as a *warmup*—and linearly decreased to $0$ afterward. We apply dropout with probability $p = 0.1$ and weight decay with $\lambda = 0.01$. We train for 3 epochs on all datasets and use global gradient clipping. Following Devlin et al. (2019), we use the AdamW optimizer (Loshchilov and Hutter, 2019) *without* bias correction.

We decided to not show results for BERT$_{\text{BASE}}$ since previous works observed no instability when fine-tuning BERT$_{\text{BASE}}$ which we also confirmed in our experiments. Instead, we show additional results on RoBERTa$_{\text{LARGE}}$ (Liu et al., 2019) and ALBERT$_{\text{LARGE-V2}}$ (Lan et al., 2020) using the same fine-tuning strategy. We note that compared to BERT, both RoBERTa and ALBERT have slightly different hyperparameters. In particular, RoBERTa uses weight decay with $\lambda = 0.1$ and no gradient clipping, and ALBERT does not use dropout. A detailed list of all default hyperparameters for all models can be found in Section 7.3 of the Appendix. Our implementation is based on HuggingFace's transformers library (Wolf et al., 2019) and is available online: `https://github.com/uds-lsv/bert-stable-fine-tuning`.

**Fine-tuning stability.** By *fine-tuning stability* we mean the standard deviation of the fine-tuning performance (measured, e.g., in terms of accuracy, MCC or $F_1$ score) over the randomness of an algorithm. We follow previous works (Phang et al., 2018; Dodge et al., 2020; Lee et al., 2020) and measure fine-tuning stability using the development sets from the GLUE benchmark. We discuss alternative notions of stability in Section 7.1 in the Appendix.

**Failed runs.** Following Dodge et al. (2020), we refer to a fine-tuning run as a *failed run* if its accuracy at the end of training is less or equal to that of a majority classifier on the respective dataset. The majority baselines for all tasks are found in Section 7.2 in the Appendix.

## 4 INVESTIGATING PREVIOUS HYPOTHESES FOR FINE-TUNING INSTABILITY

Previous works on fine-tuning predominantly state two hypotheses for what can be related to fine-tuning instability: *catastrophic forgetting* and *small training data size* of the downstream tasks. Despite the ubiquity of these hypotheses (Devlin et al., 2019; Phang et al., 2018; Dodge et al., 2020; Lee et al., 2020), we argue that none of them has a causal relationship with fine-tuning instability.

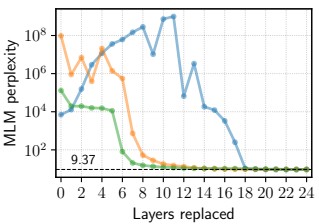 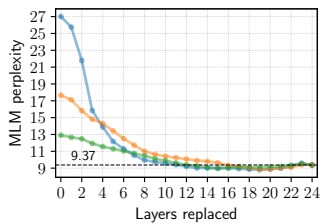 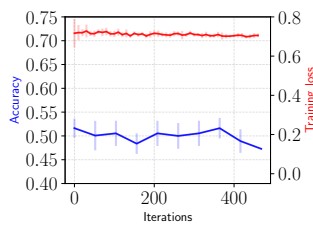

(a) Perplexity of failed models     (b) Perplexity of successful models     (c) Training of failed models

Figure 2: Language modeling perplexity for three failed (a) and successful (b) fine-tuning runs of BERT on RTE where we replace the weights of the top-$k$ layers with their pre-trained values. We can observe that it is often sufficient to reset around 10 layers out of 24 to recover back the language modeling abilities of the pre-trained model. (c) shows the average training loss and development accuracy ($\pm$1std) for 10 failed fine-tuning runs on RTE. Failed fine-tuning runs lead to a trivial training loss suggesting an optimization problem.

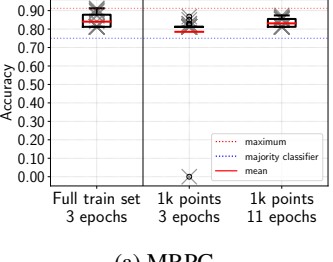 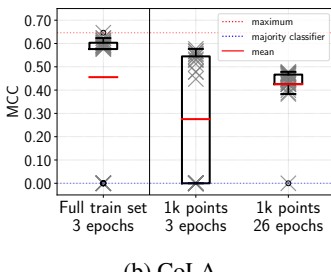 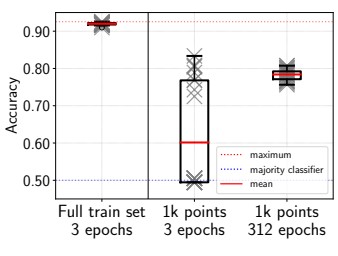

(a) MRPC             (b) CoLA             (c) QNLI

Figure 3: Development set results on down-sampled MRPC, CoLA, and QNLI using the default fine-tuning scheme of BERT (Devlin et al., 2019). The leftmost boxplot in each sub-figure shows the development accuracy when training on the full training set.

## 4.1 DOES CATASTROPHIC FORGETTING CAUSE FINE-TUNING INSTABILITY?

*Catastrophic forgetting* (McCloskey and Cohen, 1989; Kirkpatrick et al., 2017) refers to the phenomenon when a neural network is sequentially trained to perform two different tasks, and it loses its ability to perform the first task after being trained on the second. More specifically, in our setup, it means that after fine-tuning a pre-trained model, it can no longer perform the original masked language modeling task used for pre-training. This can be measured in terms of the perplexity on the original training data. Although the language modeling performance of a pre-trained model correlates with its fine-tuning accuracy (Liu et al., 2019; Lan et al., 2020), there is no clear motivation for why preserving the original masked language modeling performance after fine-tuning is important.[1]

In the context of fine-tuning BERT, Lee et al. (2020) suggest that their regularization method has an effect of alleviating catastrophic forgetting. Thus, it is important to understand how exactly catastrophic forgetting occurs during fine-tuning and how it relates to the observed fine-tuning instability. To better understand this, we perform the following experiment: we fine-tune BERT on RTE, following the default strategy by Devlin et al. (2019). We select three successful and three failed fine-tuning runs and evaluate their masked language modeling perplexity on the test set of the WikiText-2 language modeling benchmark (Merity et al., 2016).[2] We sequentially substitute the top-$k$ layers of the network varying $k$ from 0 (i.e. all layers are from the fine-tuned model) to 24 (i.e. all layers are from the pre-trained model). We show the results in Fig. 2 (a) and (b).

We can observe that although catastrophic forgetting occurs for the failed models (Fig. 2a) — perplexity on WikiText-2 is indeed degraded for $k = 0$ — the phenomenon is much more nuanced. Namely, catastrophic forgetting affects only the top layers of the network — in our experiments often around

---

[1]An exception could by the case where supervised fine-tuning is performed as an intermediate training step, e.g. with the goal of domain adaptation. We leave an investigation of this setting for future work.

[2]BERT was trained on English Wikipedia, hence WikiText-2 can be seen as a subset of its training data.

10 out of 24 layers, and the same is however also true for the successfully fine-tuned models, except for a much smaller increase in perplexity.

Another important aspect of our experiment is that catastrophic forgetting typically requires that the model at least successfully learns how to perform the new task. However, this is not the case for the failed fine-tuning runs. Not only is the development accuracy equal to that of the majority classifier, but also the training loss on the fine-tuning task (here RTE) is trivial, i.e. close to $-\ln(1/2)$ (see Fig. 2 (c)). This suggests that the observed fine-tuning failure is rather an optimization problem *causing* catastrophic forgetting in the top layers of the pre-trained model. We will show later that the optimization aspect is actually sufficient to explain most of the fine-tuning variance.

## 4.2 DO SMALL TRAINING DATASETS CAUSE FINE-TUNING INSTABILITY?

Having a small training dataset is by far the most commonly stated hypothesis for fine-tuning instability. Multiple recent works (Devlin et al., 2019; Phang et al., 2018; Lee et al., 2020; Zhu et al., 2020; Dodge et al., 2020; Pruksachatkun et al., 2020) that have observed BERT fine-tuning to be unstable relate this finding to the small number of training examples.

To test if having a small training dataset inherently leads to fine-tuning instability we perform the following experiment:[3] we randomly sample $1,000$ training samples from the CoLA, MRPC, and QNLI training datasets and fine-tune BERT using 25 different random seeds on each dataset. We compare two different settings: first, training for 3 epochs on the reduced training dataset, and second, training for the same number of *iterations* as on the full training dataset. We show the results in Fig. 3 and observe that training on less data does indeed affect the fine-tuning variance, in particular, there are many more failed runs. However, when we simply train for as many *iterations* as on the full training dataset, we almost completely recover the original variance of the fine-tuning performance. We also observe no failed runs on MRPC and QNLI and only a single failed run on CoLA which is similar to the results obtained by training on the full training set. Further, as expected, we observe that training on fewer samples affects the generalization of the model, leading to a worse development set performance on all three tasks.[4]

We conclude from this experiment, that the role of training dataset size per se is *orthogonal* to fine-tuning stability. *What is crucial is rather the number of training iterations*. As our experiment shows, the observed increase in instability when training with smaller datasets can rather be attributed to the reduction of the number of iterations (that changes the effective learning rate schedule) which, as we will show in the next section, has a crucial influence on the fine-tuning stability.

## 5 DISENTANGLING OPTIMIZATION AND GENERALIZATION IN FINE-TUNING INSTABILITY

Our findings in Section 4 detail that while both catastrophic forgetting and small size of the datasets indeed *correlate* with fine-tuning instability, none of them are causing it. In this section, we argue that the fine-tuning instability is an optimization problem, and it admits a simple solution. Additionally, we show that even though a large fraction of the fine-tuning instability can be explained by optimization, the remaining instability can be attributed to generalization issues where fine-tuning runs with the same training loss exhibit noticeable differences in the development set performance.

### 5.1 THE ROLE OF OPTIMIZATION

**Failed fine-tuning runs suffer from vanishing gradients.** We observed in Fig. 2c that the failed runs have practically constant training loss throughout the training (see Fig. 14 in the Appendix for a comparison with successful fine-tuning). In order to better understand this phenomenon, in Fig. 4 we plot the $\ell_2$ gradient norms of the loss function with respect to different layers of BERT, for one failed and successful fine-tuning run. For the failed run we see large enough gradients only for the

---

[3]We remark that a similar experiment was done in Phang et al. (2018), but with a different goal of showing that their extended pre-training procedure can improve fine-tuning stability.

[4]Section 7.8 in the Appendix shows that the same holds true for datasets from different domains than the pre-training data.

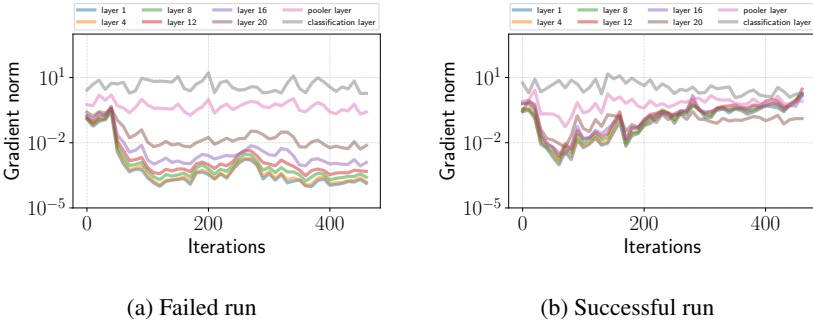

(a) Failed run ··· (b) Successful run

Figure 4: Gradient norms (plotted on a *logarithmic scale*) of different layers on RTE for a failed and successful run of BERT fine-tuning. We observe that the failed run is characterized by *vanishing gradients* in the bottom layers of the network. Additional plots for other weight matrices can be found in the Appendix.

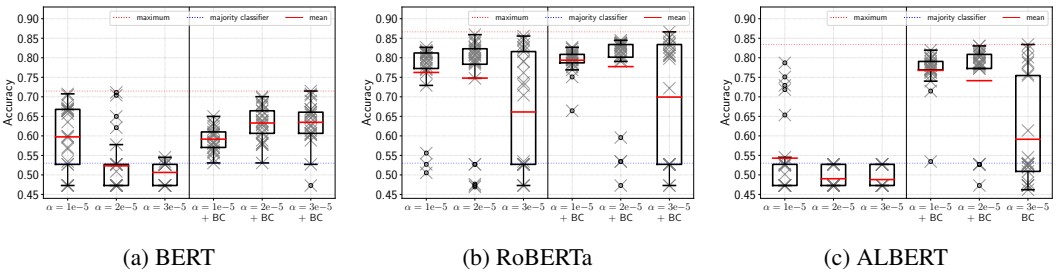

(a) BERT ··· (b) RoBERTa ··· (c) ALBERT

Figure 5: Box plots showing the fine-tuning performance of (a) BERT, (b) RoBERTa, (c) ALBERT for different learning rates $\alpha$ with and without bias correction (BC) on RTE. For BERT and ALBERT, having bias correction leads to more stable results and allows to train using larger learning rates. For RoBERTa, the effect is less pronounced but still visible.

top layers and *vanishing gradients* for the bottom layers. This is in large contrast to the successful run. While we also observe small gradients at the beginning of training (until iteration 70), gradients start to grow as training continues. Moreover, at the end of fine-tuning, we observe gradient norms nearly $2\times$ orders of magnitude larger than that of the failed run. Similar visualizations for additional layers and weights can be found in Fig. 10 in the Appendix. Moreover, we observe the same behavior also for RoBERTa and ALBERT models, and the corresponding figures can be found in the Appendix as well (Fig. 11 and 12).

Importantly, we note that the vanishing gradients we observe during fine-tuning are harder to resolve than the standard *vanishing gradient problem* (Hochreiter, 1991; Bengio et al., 1994). In particular, common weight initialization schemes (Glorot and Bengio, 2010; He et al., 2015) ensure that the pre-activations of each layer of the network have zero mean and unit variance in expectation. However, we cannot simply modify the weights of a pre-trained model on each layer to ensure this property since this would conflict with the idea of using the pre-trained weights.

**Importance of bias correction in ADAM.** Following Devlin et al. (2019), subsequent works on fine-tuning BERT-based models use the ADAM optimizer (Kingma and Ba, 2015). A subtle detail of the fine-tuning scheme of Devlin et al. (2019) is that it *does not* include the bias correction in ADAM. Kingma and Ba (2015) already describe the effect of the bias correction as to reduce the learning rate at the beginning of training. By rewriting the update equations of ADAM as follows, we can clearly see this effect of bias correction:

$$\alpha_t \leftarrow \alpha \cdot \sqrt{1 - \beta_2^t}/(1 - \beta_1^t), \tag{1}$$

$$\theta_t \leftarrow \theta_{t-1} - \alpha_t \cdot m_t/(\sqrt{v_t} + \epsilon), \tag{2}$$

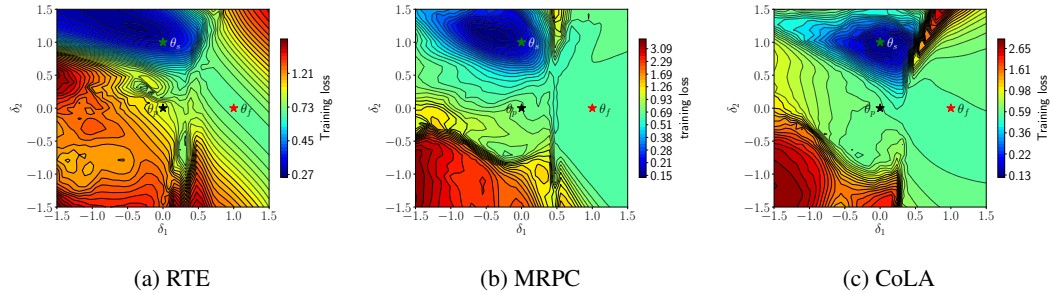

|           |           |           |
| :-------: | :-------: | :-------: |
|  (a) RTE  |  (b) MRPC |  (c) CoLA |

Figure 7: 2D loss surfaces in the subspace spanned by $\delta_1 = \theta_f - \theta_p$ and $\delta_2 = \theta_s - \theta_p$ on RTE, MRPC, and CoLA. $\theta_p, \theta_f, \theta_s$ denote the parameters of the pre-trained, failed, and successfully trained model, respectively.

Here $m_t$ and $v_t$ are biased first and second moment estimates respectively. Equation (1) shows that bias correction simply boils down to reducing the original step size $\alpha$ by a multiplicative factor $\sqrt{1 - \beta_2^t}/(1 - \beta_1^t)$ which is significantly below 1 for the first iterations of training and approaches 1 as the number of training iterations $t$ increases (see Fig. 6). Along the same lines, You et al. (2020) explicitly remark that bias correction in ADAM has a similar effect to the warmup which is widely used in deep learning to prevent divergence early in training (He et al., 2016; Goyal et al., 2017; Devlin et al., 2019; Wong et al., 2020).

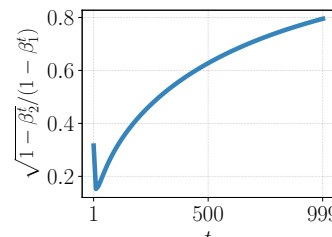

Figure 6: The bias correction term of ADAM ($\beta_1 = 0.9$ and $\beta_2 = 0.999$).

The implicit warmup of ADAM is likely to be an important factor that contributed to its success. We argue that fine-tuning BERT-based language models is not an exception. In Fig. 5 we show the results of fine-tuning on RTE with and without bias correction for BERT, RoBERTa, and ALBERT models.[5] We observe that there is a significant benefit in combining warmup with bias correction, particularly for BERT and ALBERT. Even though for RoBERTa fine-tuning is already more stable even without bias correction, adding bias correction gives an additional improvement.

Our results show that bias correction is useful if we want to get the best performance within 3 epochs, the default recommendation by Devlin et al. (2019). An alternative solution is to simply train longer with a smaller learning rate, which also leads to much more stable fine-tuning. We provide a more detailed ablation study in Appendix (Fig. 9) with analogous box plots for BERT using various learning rates, numbers of training epochs, with and without bias correction. Finally, concurrently to our work, Zhang et al. (2021) also make a similar observation about the importance of bias correction and longer training which gives further evidence to our findings.

**Loss surfaces.** To get further intuition about the fine-tuning failure, we provide loss surface visualizations (Li et al., 2018; Hao et al., 2019) of failed and successful runs when fine-tuning BERT. Denote by $\theta_p, \theta_f, \theta_s$ the parameters of the pre-trained model, failed model, and successfully trained model, respectively. We plot a two-dimensional loss surface $f(\alpha, \beta) = \mathcal{L}(\theta_p + \alpha\delta_1 + \beta\delta_2)$ in the subspace spanned by $\delta_1 = \theta_f - \theta_p$ and $\delta_2 = \theta_s - \theta_p$ centered at the weights of the pre-trained model $\theta_p$. Additional details are specified in Section 7.6 in the Appendix.

Contour plots of the loss surfaces for RTE, MRPC, and CoLA are shown in Fig. 7. They provide additional evidence to our findings on vanishing gradients: for failed fine-tuning runs gradient descent converges to a "bad" valley with a sub-optimal training loss. Moreover, this bad valley is separated from the local minimum (to which the successfully trained run converged) by a barrier (see also Fig. 13 in the Appendix). Interestingly, we observe a highly similar geometry for all three datasets providing further support for our interpretation of fine-tuning instability as a primarily optimization issue.

---

[5]Some of the hyperparameter settings lead to a small fine-tuning variance where all runs lead to a performance below the majority baseline. Obviously, such fine-tuning stability is of limited use.

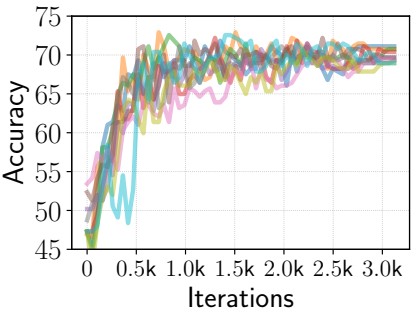

(a) Development set accuracy over training

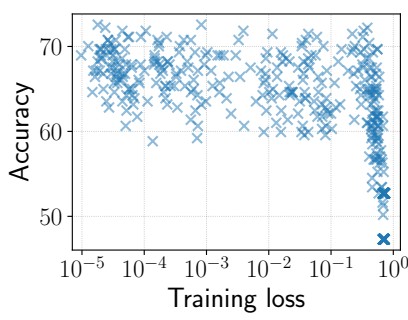

(b) Generalization performance vs. training loss

Figure 8: Development set accuracy for multiple fine-tuning runs on RTE. The models for (a) are trained with 10 different seeds, and models for (b) are taken at the end of the training, and trained with different seeds and hyperparameters.

## 5.2 THE ROLE OF GENERALIZATION

We now turn to the generalization aspects of fine-tuning instability. In order to show that the remaining fine-tuning variance on the development set can be attributed to generalization, we perform the following experiment: we fine-tune BERT on RTE for 20 epochs and show the development set accuracy for 10 successful runs in Fig. 8a. Further, we show in Fig. 8b the development set accuracy vs. training loss of all BERT models fine-tuned on RTE for the full ablation study (shown in Fig. 9 in the Appendix), in total 450 models.

We find that despite achieving close to zero training loss overfitting is not an issue during fine-tuning. This is consistent with previous work (Hao et al., 2019), which arrived at a similar conclusion. Based on our results, we argue that it is even desirable to train for a larger number of iterations since the development accuracy varies considerably during fine-tuning and it does not degrade even when the training loss is as low as $10^{-5}$.

Combining these findings with results from the previous section, we conclude that the fine-tuning instability can be decomposed into two aspects: optimization and generalization. In the next section, we propose a simple solution addressing both issues.

## 6 A SIMPLE BUT HARD-TO-BEAT BASELINE FOR FINE-TUNING BERT

As our findings in Section 5 show, the empirically observed instability of fine-tuning BERT can be attributed to vanishing gradients early in training as well as differences in generalization late in training. Given the new understanding of fine-tuning instability, we propose the following guidelines for fine-tuning transformer-based masked language models on small datasets:

- Use small learning rates with bias correction to avoid vanishing gradients early in training.
- Increase the number of iterations considerably and train to (almost) zero training loss.

This leads to the following simple baseline scheme: we fine-tune BERT using ADAM with bias correction and a learning rate of $2e{-}5$. The training is performed for 20 epochs, and the learning rate is linearly increased for the first $10\%$ of steps and linearly decayed to zero afterward. All other hyperparameters are kept unchanged. A full ablation study on RTE testing various combinations of the changed hyperparameters is presented in Section 7.4 in the Appendix.

**Results.** Despite the simplicity of our proposed fine-tuning strategy, we obtain strong empirical performance. Table 1 and Fig. 1 show the results of fine-tuning BERT on RTE, MRPC, and CoLA. We compare to the default strategy of Devlin et al. (2019) and the recent Mixout method from Lee et al. (2020). First, we observe that our method leads to a much more stable fine-tuning performance on all three datasets as evidenced by the significantly smaller standard deviation of the final performance. To further validate our claim about the fine-tuning stability, we run Levene's test (Levene, 1960) to check the equality of variances for the distributions of the final performances on each dataset. For all

Table 1: Standard deviation, mean, and maximum performance on the development set of RTE, MRPC, and CoLA when fine-tuning BERT over 25 random seeds. Standard deviation: lower is better, i.e. fine-tuning is more stable. $^\star$ denotes significant difference ($p < 0.001$) when compared to the second smallest standard deviation.

| Approach | RTE | | | MRPC | | | CoLA | | |
|---|---|---|---|---|---|---|---|---|---|
| | std | mean | max | std | mean | max | std | mean | max |
| Devlin et al. (2019) | 4.5 | 50.9 | 67.5 | 3.9 | 84.0 | 91.2 | 25.6 | 45.6 | 64.6 |
| Lee et al. (2020) | 7.9 | 65.3 | **74.4** | 3.8 | 87.8 | **91.8** | 20.9 | 51.9 | 64.0 |
| Ours | **2.7**$^\star$ | **67.3** | 71.1 | **0.8**$^\star$ | **90.3** | 91.7 | **1.8**$^\star$ | **62.1** | **65.3** |

three datasets, the test results in a p-value less than 0.001 when we compare the variances between our method and the method achieving the second smallest variance. Second, we also observe that our method improves the overall fine-tuning performance: in Table 1 we achieve a higher mean value on all datasets and also comparable or better maximum performance on MRPC and CoLA respectively.

Finally, we note that we suggest to increase the number of fine-tuning iterations only for small datasets, and thus the increased computational cost of our proposed scheme is not a problem in practice. Moreover, we think that overall our findings lead to *more efficient* fine-tuning because of the significantly improved stability which effectively reduces the number of necessary fine-tuning runs.

## 7 CONCLUSIONS

In this work, we have discussed the existing hypotheses regarding the reasons behind fine-tuning instability and proposed a new baseline strategy for fine-tuning that leads to significantly improved fine-tuning stability and overall improved results on commonly used datasets from the GLUE benchmark.

By analyzing failed fine-tuning runs, we find that neither catastrophic forgetting nor small dataset sizes sufficiently explain fine-tuning instability. Instead, our analysis reveals that fine-tuning instability can be characterized by two distinct problems: (1) optimization difficulties early in training, characterized by vanishing gradients, and (2) differences in generalization, characterized by a large variance of development set accuracy for runs with almost equivalent training performance.

Based on our analysis, we propose a simple but strong baseline strategy for fine-tuning BERT which outperforms the previous works in terms of fine-tuning stability and overall performance.

## ACKNOWLEDGMENTS

We thank Anna Khokhlova for her help with the language modeling experiments, Cheolhyoung Lee and Jesse Dodge for providing us with details of their works, and Badr Abdullah and Aditya Mogadala for their helpful comments on a draft of this paper.

This work was funded by the Deutsche Forschungsgemeinschaft (DFG, German Research Foundation) – Project-ID 232722074 – SFB 1102.

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

APPENDIX

## 7.1 ALTERNATIVE NOTIONS OF STABILITY

Here, we elaborate on other possible definitions of fine-tuning stability. The definition that we use throughout the paper follows the previous work (Phang et al., 2018; Dodge et al., 2020; Lee et al., 2020). For example, while Dodge et al. (2020) do not directly define fine-tuning stability, they report and analyze the *standard deviation* of the validation performance (e.g., see Section 4.1 of their paper). Along the same lines, an earlier work of Phang et al. (2018), which studies the influence of intermediate fine-tuning, discusses the *variance* of the validation performance (see Section 4: Results, paragraph Fine-Tuning Stability therein) and shows the *standard deviation* over multiple random seeds in Figure 1.

For simplicity, let us assume that the performance metric is *accuracy* and we have two classes. Let $A$ be a randomized fine-tuning algorithm that produces a classifier $f_A$, and let us denote data points as $(x, y) \sim \mathcal{D}$ where $\mathcal{D}$ is the data-generating distribution. Our definition of fine-tuning stability can be formalized as follows:

$$S_{\text{ours}}(A) = \text{Var}_A \left[ \text{E}_{x,y} \left[ \mathbb{1}_{f_A(x)=y} \right] \right] = \text{Var}_A \left[ \text{Accuracy}(f_A) \right].$$

This definition directly measures the variance of the performance metric and aims to answer the question: *If we perform fine-tuning multiple times, how large will the difference in performance be?*

An alternative definition of fine-tuning stability that could be considered is *per-point stability* where the expectation and variance are interchanged:

$$S_{\text{per-point}}(A) = \text{E}_{x,y} \left[ \text{Var}_A \left[ \mathbb{1}_{f_A(x)=y} \right] \right].$$

This definition captures a different notion of stability. Namely, it captures stability per data point by measuring how much the classifiers $f_A$ differ on the same point $x$ given label $y$. Studying the per-point fine-tuning stability can be useful to better understand the properties of fine-tuned models and we refer to McCoy et al. (2020) for a study in this direction.

## 7.2 TASK STATISTICS

Statistics for each of the datasets studied in this paper are shown in Table 2. All datasets are publicly available. The GLUE datasets can be downloaded here: `https://github.com/nyu-mll/jiant`. SciTail is available at `https://github.com/allenai/scitail`.

Table 2: Dataset statistics and majority baselines.

|  | RTE | MRPC | CoLA | QNLI | SciTail |
|---|---|---|---|---|---|
| Training | 2491 | 3669 | 8551 | 104744 | 23596 |
| Development | 278 | 409 | 1043 | 5464 | 1304 |
| Majority baseline | 0.53 | 0.75 | 0.0 | 0.50 | 0.63 |
| Metric | Acc. | F1 score | MCC | Acc. | Acc. |

## 7.3 HYPERPARAMETERS

Hyperparameters for BERT, RoBERTa, and ALBERT used for all our experiments are shown in Table 3.

## 7.4 ABLATION STUDIES

Figure 9 shows the results of fine-tuning on RTE with different combinations of learning rate, number of training epochs, and bias correction. We make the following observations:

- When training for only 3 epochs, disabling bias correction clearly hurts performance.
- With bias correction, training with larger learning rates is possible.
- Combining the usage of bias correction with training for more epochs leads to the best performance.

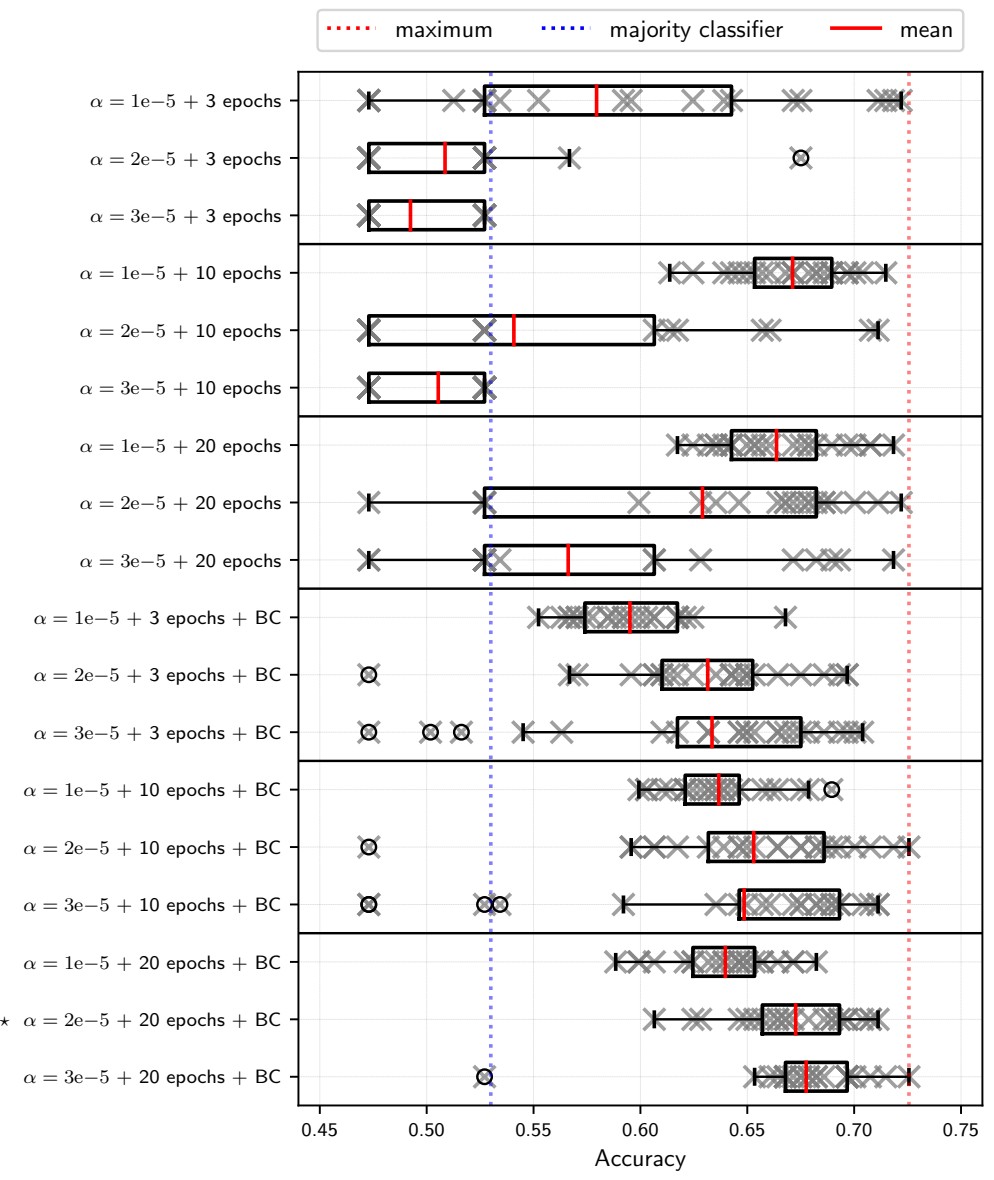

Figure 9: Full ablation of fine-tuning BERT on RTE. For each setting, we vary only the number of training steps, learning rate, and usage of bias correction (BC). All other hyperparameters are unchanged. We fine-tune 25 models for each setting. ⋆ shows the setting which we recommend as a new baseline fine-tuning strategy.

Table 3: Hyperparameters used for fine-tuning.

| Hyperparam | BERT | RoBERTa | ALBERT |
|---|---|---|---|
| Epochs | $3, 10, 20$ | $3$ | $3$ |
| Learning rate | $1e{-}5 - 5e{-}5$ | $1e{-}5 - 3e{-}5$ | $1e{-}5 - 3e{-}5$ |
| Learning rate schedule | warmup-linear | warmup-linear | warmup-linear |
| Warmup ratio | 0.1 | 0.1 | 0.1 |
| Batch size | 16 | 16 | 16 |
| Adam $\epsilon$ | $1e{-}6$ | $1e{-}6$ | $1e{-}6$ |
| Adam $\beta_1$ | 0.9 | 0.9 | 0.9 |
| Adam $\beta_2$ | 0.999 | 0.98 | 0.999 |
| Adam bias correction | {True, False} | {True, False} | {True, False} |
| Dropout | 0.1 | 0.1 | – |
| Weight decay | 0.01 | 0.1 | – |
| Clipping gradient norm | 1.0 | – | 1.0 |
| Number of random seeds | 25 | 25 | 25 |

### 7.5 ADDITIONAL GRADIENT NORM VISUALIZATIONS

We provide additional visualizations for the vanishing gradients observed when fine-tuning BERT, RoBERTa, and ALBERT in Figures 10, 11, 12. Note that for ALBERT besides the pooler and classification layers, we plot only the gradient norms of a single hidden layer (referred to as `layer0`) because of weight sharing.

**Gradient norms and MLM perplexity.** We can see from the gradient norm visualizations for BERT in Figures 4 and 10 that the gradient norm of the pooler and classification layer remains large. Hence, even though the gradients on most layers of the model vanish, we still update the weights on the top layers. In fact, this explains the large increase in MLM perplexity for the failed models which is shown in Fig. 2a. While most of the layers do not change as we continue training, the top layers of the network change dramatically.

### 7.6 LOSS SURFACES

For Fig. 7, we define the range for both $\alpha$ and $\beta$ as $[-1.5, 1.5]$ and sample 40 points for each axis. We evaluate the loss on 128 samples from the training dataset of each task using *all* model parameters, including the classification layer. We disabled dropout for generating the surface plots.

Fig. 13 shows contour plots of the total gradient norm. We can again see that the point to which the failed model converges to ($\theta_f$) is separated from the point the successful model converges to ($\theta_s$) by a barrier. Moreover, on all the three datasets we can clearly see the valley around $\theta_f$ with a small gradient norm.

### 7.7 TRAINING CURVES

Fig. 14 shows training curves for 10 successful and 10 failed fine-tuning runs on RTE. We can clearly observe that all 10 failed runs have a common pattern: throughout the training, their training loss stays close to that at initialization. This implies an optimization problem and suggests to reconsider the optimization scheme.

### 7.8 ADDITIONAL FINE-TUNING RESULTS

We report additional fine-tuning results on the SciTail dataset (Khot et al., 2018) in Table 4 to demonstrate that our findings generalize to datasets from other domains.

Due to its comparatively large size (28k training samples) fine-tuning on SciTail with the Devlin et al. (2019) scheme is already very stable even when trained for 3 epochs. This is comparable to what we find for QNLI in Section 4.1. When applying our fine-tuning scheme to SciTail, the results are very close to that of Devlin et al. (2019). On the other hand, when training on a smaller subset of SciTail

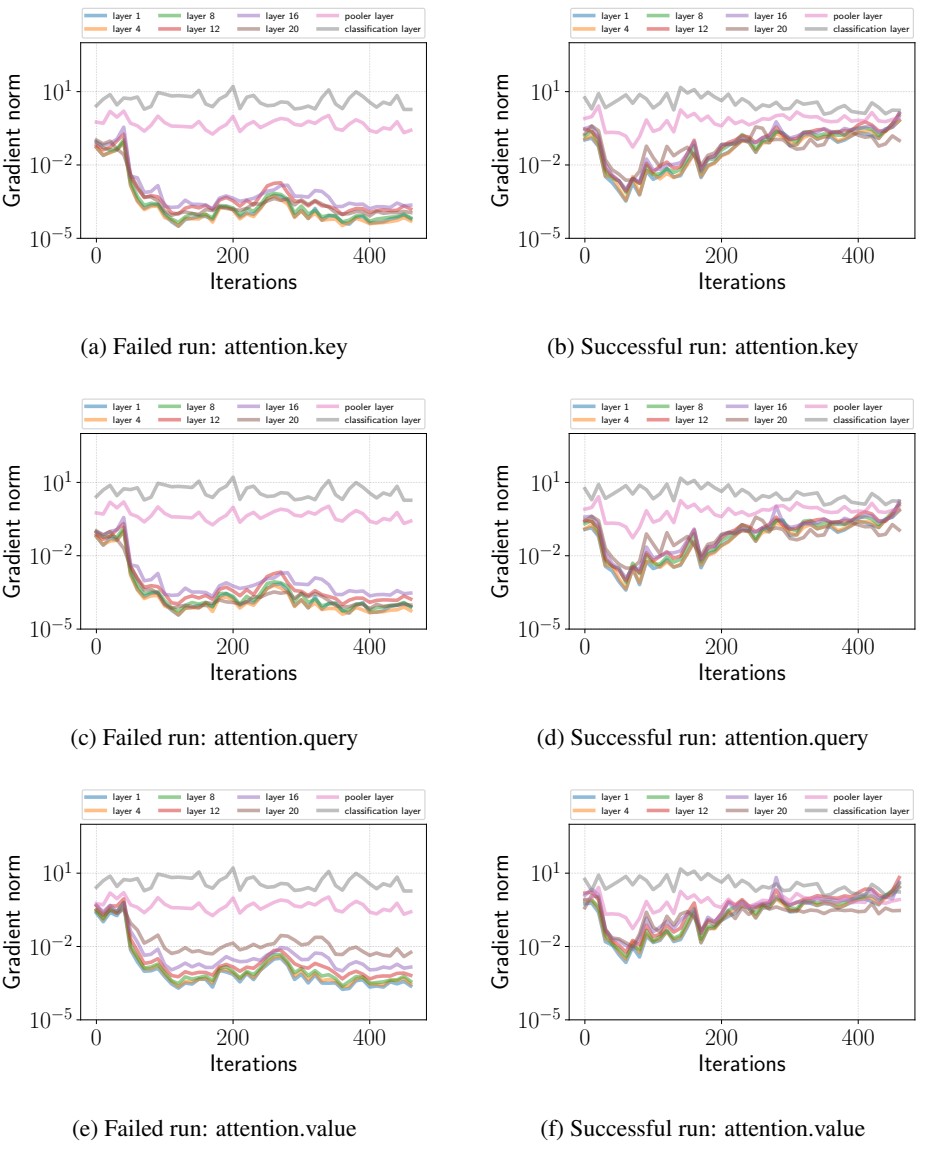

Figure 10: Gradient norms (plotted on a *logarithmic scale*) of additional weight matrices of **BERT** fine-tuned on RTE. Corresponding layer names are in the captions. We show gradient norms corresponding to a single failed and single successful, respectively.

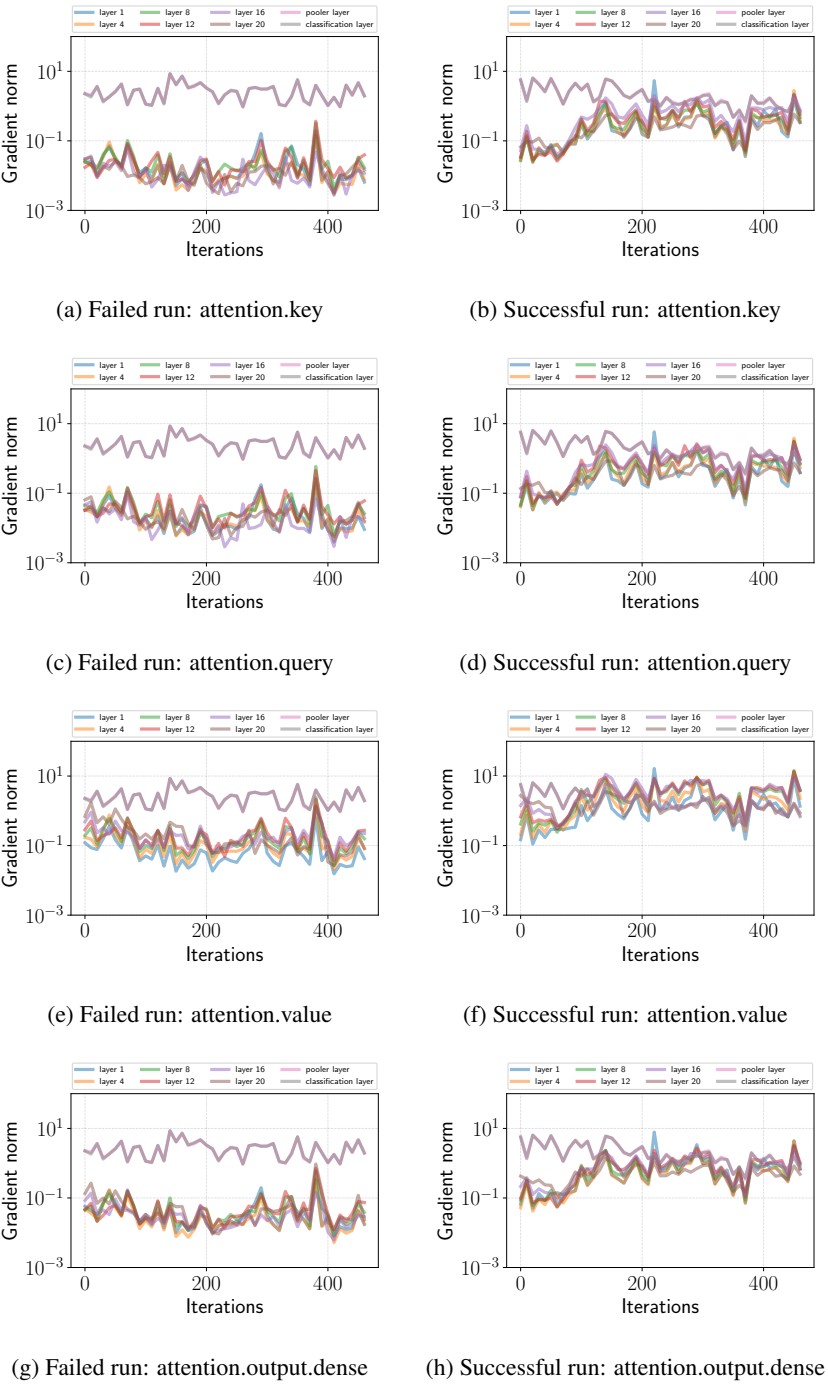

(a) Failed run: attention.key

(b) Successful run: attention.key

(c) Failed run: attention.query

(d) Successful run: attention.query

(e) Failed run: attention.value

(f) Successful run: attention.value

(g) Failed run: attention.output.dense

(h) Successful run: attention.output.dense

Figure 11: Gradient norms (plotted on a *logarithmic scale*) of additional weight matrices of **RoBERTa** fine-tuned on RTE. Corresponding layer names are in the captions. We show gradient norms corresponding to a single failed and single successful, respectively.

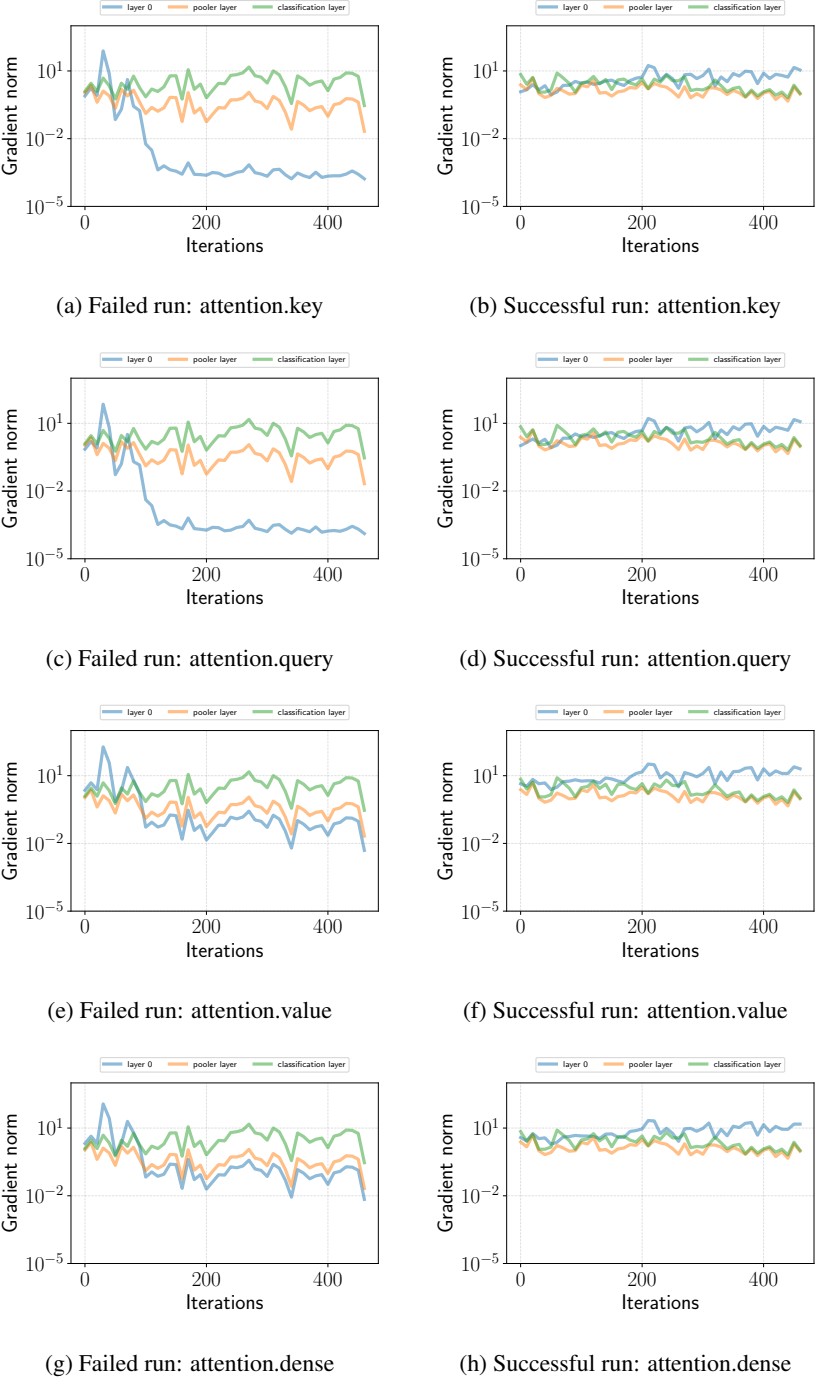

Figure 12: Gradient norms (plotted on a *logarithmic scale*) of additional weight matrices of **ALBERT** fine-tuned on RTE. Corresponding layer names are in the captions. We show gradient norms corresponding to a single failed and single successful, respectively.

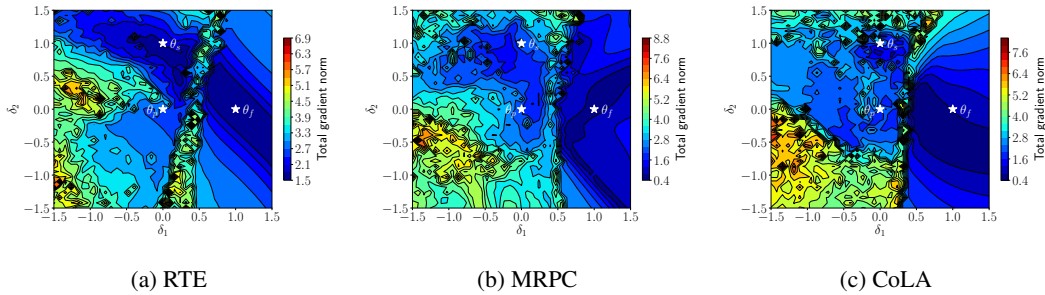

| (a) RTE | (b) MRPC | (c) CoLA |

Figure 13: 2D gradient norm surfaces in the subspace spanned by $\delta_1 = \theta_f - \theta_p$ and $\delta_2 = \theta_s - \theta_p$ for BERT fine-tuned on RTE, MRPC and CoLA. $\theta_p, \theta_f, \theta_s$ denote the parameters of the pre-trained, failed, and successfully trained model, respectively.

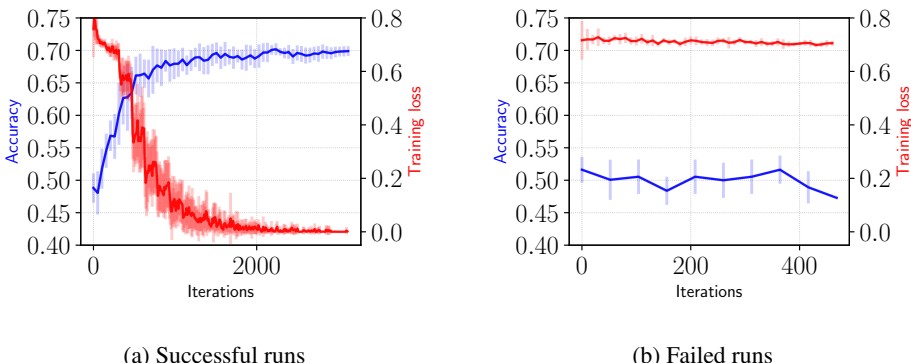

| (a) Successful runs | (b) Failed runs |

Figure 14: The test accuracy and training loss of (a) 10 successful runs with our fine-tuning scheme and (b) 10 failed runs with fine-tuning scheme Devlin on RTE. Solid line shows the mean, error bars show $\pm$1std.

Table 4: Standard deviation, mean, and maximum performance on the development set of SciTail when fine-tuning BERT over 25 random seeds. Standard deviation: lower is better, i.e. fine-tuning is more stable.

| Approach | SciTail | | |
| --- | --- | --- | --- |
| | std | mean | max |
| Devlin et al. (2019), full train set, 3 epochs | 0.4 | 95.1 | 96.0 |
| Devlin et al. (2019), 1k samples, 3 epochs | 17.9 | 69.8 | 89.4 |
| Devlin et al. (2019), 1k samples, 71 epochs | 0.9 | 87.5 | 89.1 |
| Ours, full train set | 0.6 | 95.1 | 96.8 |

(1k training samples) we can clearly see the same results as also observed in Fig. 3 for MRPC, CoLA, and QNLI, i.e. using more training iterations improves the fine-tuning stability.

We conclude from this experiment that our findings and guidelines generalize to datasets from other domains as well. This gives further evidence that the observed instability is not a property of any particular dataset but rather a result of too few training iterations based on the common fine-tuning practice of using a fixed number of epochs (and not iterations) independently of the training data size.

