# OpenReview forum: "On the Stability of Fine-tuning BERT: Misconceptions, Explanations, and Strong Baselines"
_ICLR.cc/2021/Conference — ICLR 2021 Poster_

### Official Review · AnonReviewer4 · 2020-10-28
**An interesting and potentially impactful analysis on the fine-tuning procedures of BERT and variants**

**Rating:** 6
**Confidence:** 3

**Review:**

The paper explores why fine-tuning is an unstable process, and proposes a strategy to stabilize it.

The experiments use BERT, RoBERTa, and ALBERT, fine-tuned on three popular datasets from the GLUE benchmark.

Based on the analysis of failed fine-tuning runs, the authors conclude that while catastrophic forgetting and small size of the datasets used for finetuning (both of which are common explanations for fine-tuning instability)   indeed correlate with fine-tuning instability, they do not directly cause it.

In a set of experiments it is shown that instead, fine-tuning instability is caused by: (1) optimization difficulties early in training,
characterized by vanishing gradients, and (2) differences in generalization, characterized by a large variance of development set accuracy for runs with similar training performance.

To address these issues, the authors propose the following guidelines for fine-tuning transformer-based masked language models on \emph{small} datasets:
Use small learning rates with bias correction to avoid vanishing gradients early in training; and, increase the number of iterations considerably and train to (almost) zero training loss.

Pros:
- The experimental approach is insightful and seems solid
- The paper is exceptionally well-written and illustrated
- A contribution of practical importance for a large community

Cons:
- The focus is on a particular type of problem: transformer-based masked language models that are fine-tuned on small datasets
- I noted that the proposed fine-tuning guidelines are only evaluated using BERT, whereas the failure analyses apply also to RoBERTa and AlBERT. (Does that mean that AlBERT and RoBERTa benefit less from the proposed guidelines?)

More comments:

The authors make careful word choices throughout. I'm not sure that I like this sentence though - "there is no clear motivation for why preserving the  original masked language modeling performance after fine-tuning is important". This obviously depends on the task. The end task could involve  slot-filling for example, with fine-tuning used for domain adaptation.

---

> ### Author Response · Authors · 2020-11-18
> **Response to R4**
>
> We thank Reviewer 4 for the useful feedback. We discuss below the two raised concerns and an additional comment about preserving the original pre-training performance.
>
> **1. “The focus is on a particular type of problem: transformer-based masked language models that are fine-tuned on small datasets.”**
>
> We focus on the problem of fine-tuning transformer-based masked language models because (1) it is widely used in the NLP community and (2) at the same time some aspects of it (e.g., stability) were still poorly understood. Hence, we think it is very important to obtain a better understanding of fine-tuning to be able to fix its failure cases and allow fair comparisons between different fine-tuning approaches.
>
> **2. “I noted that the proposed fine-tuning guidelines are only evaluated using BERT, whereas the failure analyses apply also to RoBERTa and ALBERT. (Does that mean that ALBERT and RoBERTa benefit less from the proposed guidelines?)”**
>
> Right, the failure analysis applies to RoBERTa and ALBERT as well, as can be seen in Figure 5. So it’s not the case that our guidelines apply *only* to BERT. They do apply to RoBERTa and ALBERT as well. Below you can find the results of fine-tuning ALBERT-large-v2 model on RTE with the default scheme of Devlin et al. (2019) and our proposed guidelines.
>
> | **Setup**      | **Mean** | **Std**     | **Max** |
> | ---        |    ----   |          ---| --- |
> | Devlin et al. (2019)      | 49.0%       | 2.5%   | 52.7% |
> | Our fine-tuning scheme     | 75.2%       | 11.4%   | 84.5% |
> | Ours fine-tuning scheme (excluding the 4 failed runs)      | 80.5%       | 1.9%   | 84.5% |
>
> As we can see from the results (and also in Figure 5.) fine-tuning ALBERT with the scheme of Devlin et al. (2019) fails for all runs. Our scheme however, results in successful fine-tuning for 21 out of 25 runs. Note that the std of our scheme is slightly larger since even with our default scheme of 20 epochs we observe 4 failed fine-tuning runs. These can, however, be easily fixed when decreasing the initial learning rate as our failure analysis in Figure. 5 shows.
>
> **3. “I'm not sure that I like this sentence though - "there is no clear motivation for why preserving the original masked language modeling performance after fine-tuning is important". This obviously depends on the task. The end task could involve slot-filling for example, with fine-tuning used for domain adaptation.”**
>
> For the setting which we study in the paper: fine-tuning for sentence-level classification we believe that our statement holds true. There is no general motivation for why we should preserve the MLM capabilities of the model.
>
> However, we agree that for other end-tasks like slot-filling, the MLM capabilities of the model are still important. We think it can also be important in some other scenarios like intermediate fine-tuning to perform domain adaptation. We will update the text accordingly to clarify our claim.

---

### Official Review · AnonReviewer2 · 2020-10-29
**Encouraging Discussion through Simplicity**

**Rating:** 6
**Confidence:** 3

**Review:**

################################

Summary:

This paper considers the stability of fine-tuning BERT-LARGE models, with considerations for RoBERTa and ALBERT. In particular, it aims to demonstrate that previously identified reasons, catastrophic forgetting and small fine-tuning datasets, fail to explain the observed instability. Instead, it posits that the instability is caused by optimization difficulties that lead to vanishing gradients.

################################

Reasons for score:

Overall, I lean toward accept. The underlying argument is simple and slightly advances a conversation about the underlying workings of BERT-based models while leaving substantial room for future exploration. This work would be strengthened by clarifying prescriptive guidance on the baseline recommended and experimentation on additional datasets.

################################

Strengths:

- Broad interest. BERT-based models have become prevalent in NLP due to their strong empirical performance. However, variation in fine-tuning hinders the consistency of the demonstrated gains. While existing hypotheses have been made, this work provides another point of evidence to advance the conversation around fine-tuning.

- Presentation. The linearity of the argument is easy to follow and stylistic choices, e.g. the motivating question in the introduction, highlight the key aspects of this work. As a reader this is appreciated insofar as claims are stated unambiguously and then can be validated by the work that follows rather than intuited.

################################

Weaknesses:

- Prescriptive baseline. The two guidelines in Section 6 are excellent, but as a reader I immediately wonder how these generalize to the datasets where I might apply BERT-based models. The guidelines of using a "small learning rate with bias correction" and "increase the number of iterations" are good, but can they be characterized in more detail? In other words, how can I make these (hopefully) generalizable insights concrete for future applications?

- Additional datasets. While the existing datasets are sufficient for illustrating the arguments made, additional datasets would further support generalizable insights by providing additional points of evidence with respect to dataset size, and variance. This may support the point above regarding a characterization of large vs. small.

################################

Questions:

- Is there any additional insight you can provide characterizing the baseline? If I were to apply these insights to a NLI dataset in another domain, e.g. MedNLI, could I expect these to hold?

- While the aspects of pretraining and continual pretraining are explicitly deferred in the related works section, are there any insights you can provide about their impact on these findings for fine-tuning?


################################

After response:

Thank you for the clarifications. The response and changes address several of my concerns. While I will keep the score currently, I would consider it slightly higher given the additional information (i.e. ~6.5).

---

> ### Author Response · Authors · 2020-11-18
> **Response to R2**
>
> We thank Reviewer 2 for the detailed feedback and encouraging comments. We discuss the two weaknesses (W1, W2) and the first question (Q1) together since they are very related, and then discuss the last question (Q2) about continual pretraining.
>
> **1. W1: Generalizable insights for the prescriptive baseline, W2: Testing additional datasets, Q1: Additional insights: If I were to apply these insights to a NLI dataset in another domain, e.g. MedNLI, could I expect these to hold?**
>
> Unfortunately, we couldn’t run experiments on the MedNLI dataset based on its quite restricted access. However, to illustrate that our findings generalize to other datasets as well, we ran additional experiments on the SciTail dataset ([Khot et al. (2018)](https://www.aaai.org/ocs/index.php/AAAI/AAAI18/paper/view/17368)). We choose SciTail because it's based on a corpus of scientific questions from school exams which makes it a different domain compared to the pre-training data used to train BERT.
>
> We run the following experiments on SciTail: 1) we repeat the downsampling experiment shown in Fig. 3 of our paper and 2) we compare fine-tuning with the scheme of Devlin et al. (2019) to our proposed scheme. The results of the accuracy over 25 fine-tuning runs are shown below:
>
> | **Setup**      | **Mean** | **Std**     | **Max** |
> | ---        |    ----   |          --- |  --- |
> | Devlin et al. (2019), full train set, 3 epochs      | 95.1%       | 0.4%   | 96.0% |
> | Devlin et al. (2019), 1k points, 3 epochs      | 69.8%       | 17.9%   | 89.4% |
> | Devlin et al. (2019), 1k points, 71 epochs     | 87.5%       | 0.9%   | 89.1% |
> | Our fine-tuning scheme, full train set     | 95.1%       | 0.6%   | 96.8% |
>
> Note that due to its comparatively large size (28k training samples) fine-tuning on SciTail with the Devlin scheme is already very stable even when trained for 3 epochs. This is comparable to what we find for QNLI in our paper.
>
> When applying *our fine-tuning scheme* to SciTail, the results are very close to that of Devlin et al. (2019). On the other hand, for the downsampling experiment we can clearly see the same results as also observed in Fig. 3 for MRPC, CoLA, and QNLI, i.e. using more training iterations improves the fine-tuning stability.
>
> We conclude from this experiment that our findings and guidelines generalize to datasets from other domains. This gives further evidence that the observed instability is not a property of *any particular* dataset but rather a result of too few training iterations based on the common fine-tuning practice of using a fixed number of *epochs* (and not *iterations*) independently of the training data size.
>
> **2. Q2: While the aspects of pretraining and continual pretraining are explicitly deferred in the related works section, are there any insights you can provide about their impact on these findings for fine-tuning?**
>
> We will focus on the supervised intermediate/continual pre-training setting in our answer.
>
> Our findings regarding failure modes and instability also apply to the intermediate task fine-tuning itself. Moreover, it should be noted that intermediate task fine-tuning is typically done on large datasets, see e.g. [Pruksachatkun et al. (2020)](https://arxiv.org/abs/2005.00628): 10 out of 11 of their intermediate fine-tuning tasks have more than 20k training samples. So we wouldn’t expect to see large instabilities here. And if there would be instabilities, our proposed scheme would likely help to avoid them.
>
> Another interesting aspect is what happens *after* the intermediate fine-tuning, i.e. how it affects the stability of fine-tuning on the downstream task. We’d like to point again to [Pruksachatkun et al. (2020)](https://arxiv.org/abs/2005.00628) in order to answer this question who find that *“intermediate-task training reduces the likelihood of degenerate runs, leading to ostensibly positive transfer results on those four acceptability judgment tasks across most intermediate tasks.”*
> And moreover: *“extremely negative transfer from intermediate-task training can also result in a higher frequency of degenerate runs in downstream tasks, as we observe in the cases of using QQP and SocialIQA as intermediate tasks."*
>
> So it seems that intermediate supervised fine-tuning can have both a positive and negative impact on downstream task fine-tuning. However, compared to our work it requires additional data. It would definitely be very interesting to study the influence of intermediate fine-tuning more closely in future work.

---

### Official Review · AnonReviewer1 · 2020-10-29
**Clear, technically correct, experimentally rigorous, reproducible investigation on instability of fine-tuning BERT.**

**Rating:** 8
**Confidence:** 4

**Review:**

What is the goal of the paper?
Investigating stability of fine-tuning BERT.

What has been done before?
Finetuning BERT exhibit a large training instability (Devlin et al., 2019; Dodge et al., 2020) i.e. training the same model with multiple random seeds can result in a large variance of the task performance.

Few methods have been proposed to solve the observed instability (Phang et al., 2018; Lee et al., 2020), however without providing a sufficient understanding of why fine-tuning is prone to such failure.

This work tries to answer the question : Why is fine-tuning prone to failures and how can we improve its stability?

Another line of work investigates optimization difficulties of pre-training transformer-based language models (Xiong et al., 2020; Liu et al., 2020). Both works focus on pre-training and thus orthogonal to this work.

What are the contributions of the paper?
Investigating two common hypotheses for fine-tuning instability: catastrophic forgetting and small size of the fine-tuning datasets and demonstrating that both hypotheses fail to explain fine-tuning instability.

Investigating fine-tuning failures on datasets from the popular GLUE benchmark and show that the observed fine-tuning instability can be decomposed into two separate aspects: (1) optimization difficulties, characterized by vanishing gradients, and (2) differences in generalization, characterized by a large variance of development set accuracy for runs with almost equivalent training loss.

Presented a simple but strong baseline that makes fine-tuning BERT-based models significantly more stable than the previously proposed approaches.


What are the key techniques/experiments used to investigate this task?
To investigate catastrophic forgetting - comparing language modeling perplexity for failed and successful fine-tuning runs of BERT.
To investigate the effect of small size of the fine-tuning datasets - compare fine-tuning using  downsampled data sets for 3 epochs (standard) vs. more epochs.
Visualizing gradient norms of different layers for a failed and successful run of BERT fine-tuning.
Loss surface visualizations of failed and successful runs when fine-tuning BERT
Visualizing development accuracy vs. training loss at the end of the training for all BERT models fine-tuned for the paper
Fine-tuning performance of (a) BERT, (b) RoBERTa, (c) ALBERT for different learning rates α with and without bias correction (BC)


What are the main results?
Does catastrophic forgetting cause fine-tuning instability? No.
Catastrophic forgetting occurs for both failed and successful models in the top layers of the network,  except for a much smaller increase in perplexity in case of successful models. Catastrophic forgetting typically requires that the model at least successfully learns how to perform the new task. However, this is not the case for the failed fine-tuning runs.

Do small training datasets cause fine-tuning instability? No.
Training on less data does indeed affect the fine-tuning variance. However, when one simply trains for as many iterations as on the full training dataset, one almost completely recovers the original variance of the fine-tuning performance.

Observed instability is caused by
optimization difficulties that lead to vanishing gradient : Development accuracy of failed fine-tuning runs is less or equal to that of the majority classifier, but also the training loss on the fine-tuning task is trivial. This suggests that the observed fine-tuning failure is rather an optimization problem causing catastrophic forgetting in the top layers of the pre-trained model.
differences in generalization : Training on fewer samples affects the generalization of the model, leading to a worse development set performance on all three tasks. The role of training dataset size per se is orthogonal to fine-tuning stability. What is crucial is rather the number of training iterations.

A simple but hard to beat baseline for fine-tuning bert (better standard deviation, better mean, and competitive maximum performance)
• Use small learning rates with bias correction to avoid vanishing gradients early in training.
• Increase the number of iterations considerably and train to (almost) zero training loss.

Strengths
The approach is well motivated and well-placed in the literature.
Paper claims look correct technically and are experimentally rigorous.
Paper is easy and clear to read.
Authors have made an attempt to make their findings reproducible.
Findings apply not only to the widely used BERT model but also to more recent pre-trained models such as RoBERTa and ALBERT. These findings should benefit others as fine tuning BERT based models is a very common practice now.


Weaknesses
The role of generalization has not been discussed rigorously.
Different sets of GLUE datasets were used for different experiments without any explanation for the selection.

---

> ### Author Response · Authors · 2020-11-18
> **Response to R1**
>
> We thank Reviewer 1 for the very detailed and positive feedback. We clarify the two raised concerns below.
>
> **1. ”The role of generalization has not been discussed rigorously.“**
>
> We decided to not focus too much on the generalization aspect since we have already identified *optimization* as the main source of the fine-tuning instability. At the same time, we wanted to highlight that there is still some contribution of *generalization*, although it is much less important compared to that of the optimization aspect. We leave a more detailed investigation of the generalization aspect for future work.
>
> **2. ”Different sets of GLUE datasets were used for different experiments without any explanation for the selection.”**
>
> We focus on the RTE, MRPC, and CoLA datasets for the sake of comparison with previous work, e.g. [Dodge et al. (2020)](https://arxiv.org/abs/2002.06305) and [Lee et al. (2020)](https://arxiv.org/abs/1909.11299). We additionally run experiments on QNLI because of its large number of training samples, but alternatively we could have selected any other dataset of comparable size.  Also, we would like to highlight that based on our experiments, the observed instability is not a property of any *particular dataset* but rather a result of too few training iterations and the common fine-tuning practice of using a fixed number of *epochs* (and not *iterations*) independently of the training data size.
>
> Additionally, we think our response to R2 can also be of interest in the context of this question, where we provide additional results on the SciTail dataset.

---

### Official Review · AnonReviewer3 · 2020-10-30
**Reviews and Comments**

**Rating:** 4
**Confidence:** 5

**Review:**

### Overview

The paper focuses on the instability phenomenon happening in the fine-tuning of BERT-like models in downstream tasks. The reasons of such instability were assumed to be catastrophic forgetting and the small size of datasets on which being fine-tuned in previous literature. The authors conduct experiments on several sub tasks of GLUE in an attempt to show the aforementioned two assumptions cannot explain the instability of fine-tuning. Instead they claim that the real reasons are gradient vanishing and the lack of generalization and subsequently propose a set of training hyperparameters to improve the stability.

### Pros

The analysis of the impact of dataset size is designed clear and concise, which shows it is the number of iterations that really matters rather than the size of datasets. This conclusion is valuable for the audience.

The suggestions of training hyperparameters provided at the end of the paper is somewhat useful for future researchers.

### Cons

The authors define term "stability" by "the std of fine-tuning performance (acc, F1, MCC), the lower the better". This metric is too curtness to precisely represent the real stability. A very simple counter example: assume a fine-tuned model A has 50% accuracy on a binary classification task while model B's predictions are all opposite to model A's predictions, which also makes it a 50% accuracy model. These two models obviously show the most instable situation that they disagree on every single sample. While the std of performance is 0, which by the definition of the authors it is the most stable case. The correct practice is to measure how much it changes on the prediction of the same input between different models.

Since the definition is very important and used through the whole analysis, the fact that it is defined too rough makes the following conclusions in the paper not convincible enough.

Besides, the proposed improvement method is only using slightly different training hyperparameters, which is possible to be covered by a simple grid-search of hyperparameters and therefore cannot be seen as a big contribution.

---

> ### Author Response · Authors · 2020-11-18
> **Response to R3**
>
> We thank Reviewer 3 for the feedback. We would like to clarify in detail the main concern about the definition of fine-tuning stability that we use throughout the paper.
>
> **1. Our definition of fine-tuning stability follows the previous work.**
> * While [Dodge et al. (2020)](https://arxiv.org/abs/2002.06305) do not directly define fine-tuning stability, they report and analyze the *standard deviation* of the validation performance (e.g., see Section 4.1 of their paper).
> * Along the same lines, an earlier work of [Phang et al. (2018)](https://arxiv.org/abs/1811.01088) which studies the influence of intermediate fine-tuning, discusses the *variance* of the validation performance (see Section 4: Results, paragraph Fine-Tuning Stability therein) and plots the *standard deviation* over multiple random seeds in Figure 1.
>
>
> **2. The suggested definition of stability serves a different goal compared to what we aim to study.**
> We would like to explain the main difference between our definition of stability and the suggested one. For simplicity, let’s assume that the performance metric is *accuracy* and we have two classes (as, e.g., in RTE dataset). Let $A$ be a randomized fine-tuning algorithm that produces a classifier $f_A$, and denote data points as $(x, y) \sim \mathcal{D}$ where $\mathcal{D}$ is the data-generating distribution.
> - **Our definition of stability:**
> $S_{ours}(A) = \text{Var}_A \left[ \text{E}_\{x, y\} \left[ \mathbb{1}_\{ f_A(x) = y\} \right] \right] = \text{Var}_A \left[ \text{Accuracy}(f_A)\right],$
> This definition directly measures the variance of the performance metric and aims to answer the question: *If we perform fine-tuning multiple times, how large the difference in the performance will be?* This is precisely the measure of stability we are interested in.
> - **Suggested definition of stability:**
> We believe that the suggested definition can be formalized as follows:
> $S_{suggested}(A) = \text{E}_\{x, y\} \left[ \text{Var}_A \left[ \mathbb{1}_\{ f_A(x) = y\} \right] \right].$
> This definition also captures some notion of stability but different compared to the one we aim to study. Namely, it captures *per-point stability* by measuring how much the classifiers $f_A$ differ on the same point $x$ with respect to label $y$. Considering this question is *not* the goal of our study and the provided example from our perspective still leads to stable fine-tuning:
> "*assume a fine-tuned model A has 50% accuracy on a binary classification task while model B's predictions are all opposite to model A's predictions, which also makes it a 50% accuracy model*"
> Such stability is obviously not very useful as we mention in Footnote 3 in our paper: *Some of the hyperparameter settings lead to a small fine-tuning variance where all runs lead to a performance below the majority baseline. Obviously, such fine-tuning stability is of limited use.*
>
>
> **3. “Besides, the proposed improvement method is only using slightly different training hyperparameters”**
>
> We see the simplicity of the proposed method rather as an advantage. We think that if the stability problem can be addressed simply by changing some hyperparameters of the standard training scheme, then it can serve as a good baseline before introducing more complex methods.
>
> Moreover, we would like to emphasize that it is impractical to do a hyperparameter search over all possible hyperparameters as there are too many of them in a BERT-scale model. Particularly, because some hyperparameters like the bias correction of ADAM seem to have no influence in most cases so they are usually kept fixed. This highlights the importance of our analysis part in Section 5, which we view as an important contribution of our paper, where we first identified the problem and then proposed a solution *based on our analysis*.

---

### Decision · Program_Chairs · 2021-01-07
**Final Decision**

**Decision:**

Accept (Poster)

**Comment:**

This paper identifies the causal factors behind a major known issue in deep learning for NLP: Fine-tuning models on small datasets after self-supervised pretraining can be extremely unstable, with models needing dozens of restarts to achieve acceptable performance in some cases. The paper then introduces a simple suggested fix.

Pros:
- The motivating problem is important: A large fraction of all computing time used on language-understanding tasks involves fine-tuning runs under the protocol studied here, and the problem of fine-tuning self-supervised models should be of broader interest at ICLR.
- The proposed fix is simple and well-demonstrated. It consists of only an adjustment to the range of values considered in hyperparameter tuning (which is significant, since BERT and related papers *explicitly advise* users to use inappropriate values) and an adjustment to the implementation of the optimizer.

Cons:
- The method is demonstrated on a relatively small set of difficult text-classification datasets, so the behavior studied here may be different in very different dataset size, task difficulty, or label entropy regimes.

This paper was divisive, so I gave it a fairly close look myself, and I'm persuaded by R1 and the other two positive reviewers: This is a classic example of a 'strong baselines paper', in that demonstrates that a more careful use of established methods can obviate the need for additional tricks.

R3 raised two major concerns that they presented as potentially fatal, but that I find unpersuasive.
- This paper studies stability in model performance, not stability in predictions on individual data points. R3 argues that the latter sense of stability is the more important problem. Stability is an ambiguous term in this context, and both versions of the problem are interesting. However, as the authors pointed out, the definition of stability that is used here is consistent with previous work, and is widely accepted to be a major practical problem in NLP. I don't think this is a weakness of this paper, rather, it's an opportunity for someone else to write another, different paper on a different problem.
- R3 claims that the results are described as being more positive than they actually are, and the figure is potentially misleading. Looking at the quantitative results with R3's points in mind, I still see clear support for both of the paper's main suggestions. R3 opened up some potentially important questions about the handling of outliers in particular, but these questions were raised too late for the authors to be allowed to respond, and I don't see any evidence in the paper that anything improper was done. The marked outliers are clearly much farther from the mean/median in terms of standard deviations than the unmarked outliers. So, I don't see any evidence that these concerns reflect real methodological problems.

---

> ### Author Response · Authors · 2021-01-18
> **Response to Final Decision**
>
> Dear AC,
>
> Thank you for your time to read our paper in detail and write a very thorough meta-review.
>
> However, based on the current review from R3, we don't quite understand what is the second "major concern" exactly. Apparently, there has been a further discussion between reviewers and AC and some statements mentioned in the meta-review are quite surprising for us to read:
> - "R3 claims that the results are described as being more positive than they actually are"
> - "the figure is potentially misleading"
> - "R3 opened up some potentially important questions about the handling of outliers"
>
> We don't understand at the moment what can be a problem with our figures since we use standard boxplots. We opted for this presentation of the fine-tuning results specifically to show the whole distribution (25 points taken over 25 random seeds) of the performance metric without hiding any information about outliers.
>
> Thus, we would like to kindly ask about clarifications of R3's concerns so that we can address them in the camera-ready version. In particular, we would be happy to change our boxplots in case there is a better alternative to show the distribution of the results.
>
> Thank you.

---

> > ### Comment · Area_Chair1 · 2021-01-18
> > **Nothing to see here, but inviting R3 to comment**
> >
> > For the record: I acknowledge that it was probably a mistake for me to allude to outliers in the public metareview—R3's most recent comments were private (they were made during a period when all comments were required to be private), so you haven't had a chance to reply, and I'm not sure that I can or should paraphrase them myself. As I say at the end of that paragraph, I'm not convinced that there's anything unusual or problematic about the presentation of results.
> >
> > So, I don't think there's an issue here—the paper looks sound. R3, if you disagree, and you'd like to follow up with public suggestions, go for it.